# Microtubules as Regulators of Neural Network Shape and Function: Focus on Excitability, Plasticity and Memory

**DOI:** 10.3390/cells11060923

**Published:** 2022-03-08

**Authors:** Fernando Peña-Ortega, Ángel Abdiel Robles-Gómez, Lorena Xolalpa-Cueva

**Affiliations:** Departamento de Neurobiología del Desarrollo y Neurofisiología, Instituto de Neurobiología, Universidad Nacional Autónoma de México, Campus Juriquilla, Boulevard Juriquilla 3001, Querétaro 76230, Mexico; angelrg94@gmail.com (Á.A.R.-G.); lorena_xolalpa@outlook.es (L.X.-C.)

**Keywords:** microtubules, protein Tau, excitability, synaptic plasticity, memory

## Abstract

Neuronal microtubules (MTs) are complex cytoskeletal protein arrays that undergo activity-dependent changes in their structure and function as a response to physiological demands throughout the lifespan of neurons. Many factors shape the allostatic dynamics of MTs and tubulin dimers in the cytosolic microenvironment, such as protein–protein interactions and activity-dependent shifts in these interactions that are responsible for their plastic capabilities. Recently, several findings have reinforced the role of MTs in behavioral and cognitive processes in normal and pathological conditions. In this review, we summarize the bidirectional relationships between MTs dynamics, neuronal processes, and brain and behavioral states. The outcomes of manipulating the dynamicity of MTs by genetic or pharmacological approaches on neuronal morphology, intrinsic and synaptic excitability, the state of the network, and behaviors are heterogeneous. We discuss the critical position of MTs as responders and adaptative elements of basic neuronal function whose impact on brain function is not fully understood, and we highlight the dilemma of artificially modulating MT dynamics for therapeutic purposes.

## 1. Introduction

Neurons are postmitotic, highly polarized cells with complex morphological and functional compartments, such as soma, axon, dendrites, and synapses [1] (Figure 1). These compartments are supported morphologically and functionally on specialized and specific cytoskeletal arrangements [2,3,4,5] (Figure 1). The cytoskeleton comprises filamentous actin, intermediate filaments, microtubules (MTs), and associated proteins [6,7]. MTs are particularly important because they support the neuronal complex and dynamic branching and compartmentalization while acting as an intracellular roadmap for protein motors to deliver important cargoes (e.g., receptors, neurotransmitters) and organelles to various cell regions [1] and elements that modulate (and are modulated by) fundamental neural functions [1,8], as will be discussed in this review.

MTs are dynamic, cylindrical, linear, and anionic polymers with a hollow tubular structure. They have variable lengths [9] and are composed of α- and β-tubulin, which interact noncovalently to form a stable heterodimer with polarity orientation [10,11], a fast-growing plus end, and a slow-growing minus end [12]. MT assembly requires α- and β-tubulin to be bound to GTP, which is permanently bound to α-tubulin [13,14] and hydrolyzed to GDP by β-tubulin after tubulin dimers assembly into the MT [15,16]. The GDP attached to β-tubulin incorporated into MTs can only be replaced by GTP when tubulin returns to its heterodimeric form during the continuous cycles of polymerization and depolymerization [9].

MTs in differentiated neurons are not attached to the centrosome [17,18], which allows them to exist in different neuronal compartments as autonomous structures [19,20]. In neurons, most axonal MTs have their plus ends directed away from the cell body [21]. Their cargo transport is controlled in a directional manner and has kinesin motors carrying cargo toward axon terminals (anterograde) and dynein motors carrying cargo away from axon tips (retrograde) [22]. As will be reviewed later, MT localization at the presynaptic terminals has been recently accepted [23,24] (Figure 1). In contrast to axons, dendrites have a mixed MT orientation with both minus and plus ends oriented toward the cell body [20,25]. Although a fraction of the axonal MT array is indeed quite stable, a fraction close to the terminal is not, with dendritic MTs having a lower fraction of stable MT polymers than axons [26,27].

Neurons possess stable and dynamic pools of MTs that exhibit complex behavior, with some MTs in a growing phase, others stationary, and some in a state of disassembly [7]. The stochastic switch between growth to shortening (named catastrophe) and shortening to growth (named rescue) precludes MTs from reaching a steady-state length and makes them a structure that simultaneously undergoes assembly and disassembly, producing a condition named “dynamic instability” [28,29,30]. Dynamic instability is controlled by GTP hydrolysis and, thus, is an energy-consuming process [31] that varies depending on the isoforms of α- and β-tubulin incorporated into the MTs, their posttranslational modifications, and their interaction with MT associated proteins (MAPs) [32,33]. Despite being constantly changing structures, MTs have sufficient longevity to be substrates for tubulin modifying enzymes that lead to their tyrosination, detyrosination, acetylation, D2 modification, glutamylation, glycation, palmitoylation, and phosphorylation [34,35,36,37,38]. These posttranslational modifications modulate their binding to particular MAPs, motor proteins, or proteases [39,40,41,42], which are subject to these same posttranslational regulations.

MAPs constitute a class of heterogeneous regulators that modulate MT stability and dynamics, the directional transport of cargo, and MT nucleation. They also dynamically interact with other cellular proteins and organelles [43,44]. The brain expresses several MAPs such as Tau, MAP1A, MAP1B, MAP2, and MAP6, with some exhibiting a compartment-specific distribution. For example, MAP2A and MAP2B are preferentially located in the cell bodies and dendrites of mature neurons [45], whereas Tau protein and MAP6 are distributed mostly in the axonal compartment [46] and, to a lesser extent, in dendrites [47,48]. MAPs play a major role in the induction of distinctive morphologies between axons and dendrites [49,50], but they also regulate axonal transport [51] and neuronal plasticity [52] (Figure 1).

As will be reviewed here, MTs are fundamental in a variety of neural functions including neuronal excitability, synaptic coupling, synaptic plasticity, and memory, which are strongly dependent on a constant shift between MT stability and instability [53,54,55,56,57,58,59]. Thus, neurons are particularly susceptible to MT deregulation and defects closely related to neurological disorders [60,61,62]. MT destabilization has been associated to amyotrophic lateral sclerosis [63,64], Parkinson’s disease [65], Huntington’s disease [66], frontotemporal dementia with parkinsonism linked to chromosome 17 [67], Alzheimer’s disease (AD) [68] or psychiatric diseases [34,69,70,71,72] (Figure 1). Moreover, the relevance of MTs in brain function has also been reflected in the neuropathy induced by several MT stabilizers when used as anticancer drugs [73,74]. All these phenomena indicate that there is a delicate balance in MT stability/instability that is required for proper brain function [57]. Thus, the purpose of this review is to reveal the role of MTs and their endogenous and exogenous stability modulators in neural function and to shape the impact of neural function on MT configuration.

## 2. Modulation of MT Stability and Its Impact on Brain Function

Fibroblast growth factor 13 (FGF13), is an endogenous MT stabilizer. It is a non-secretory protein of the FGF family that acts intracellularly as an MT-stabilizing protein by promoting tubulin polymerization; its deletion leads to alterations in learning and memory [75]. As mentioned, MAPs bind and stabilize MTs in a phosphorylation-dependent manner [45,46,76] and their alterations disrupt brain function. For instance, the deletion of MAP6 triggers various neurotransmission and behavioral defects, leading to schizoaffective disorder, which could be corrected by the pharmacological stabilization of MTs [77]. The most prominent MAP is tau, which is encoded by the MAPT (microtubule-associated protein tau) gene [78], which generates six isoforms through alternative splicing [79]. Tau binds tubulin in a phosphorylation-dependent manner via its MT-binding domains [80], with a single Tau molecule crosslinking multiple tubulin dimers [81], which stabilizes MTs [82]. Although Tau is preferentially expressed within the axon, increasing its concentration towards the distal end [27], it is also located in the soma and dendrites [4,5,49,83,84]. Phosphorylation of tau, which is tightly regulated under physiological conditions [85,86], reduces its affinity for MTs, promoting their dismantling [87]. Tau hyperphosphorylation leads to aggregation and MT disruption [88,89,90,91], which is mainly due to its reduced MT-stabilizing properties but also to the sequestration of other MAPs [69,91], ultimately inducing neuronal dysfunction [92,93]. However, as will be reviewed later, blocking Tau phosphorylation also leads to neuronal morphological and functional alteration [5,94], which indicates that Tau phosphorylation has specific physiological functions, such as the modulation of N-methyl-D-aspartate (NMDA)-mediated processes [5,84,95]. Research on the role of MTs and associated proteins in brain function has paved the way for the use of pharmacological agents that modulate their stability. These agents have revealed the functional consequences of changes in MT stability under normal and pathological conditions.

## 3. Pharmacological Modulators of MT Stability

Microtubule stabilizing agents (MSAs), are among the most clinically used chemotherapeutic drugs [96] because they can inhibit cell division by stabilizing MTs [97,98]. However, MSAs have also been considered potential candidates for the treatment of neurological alterations related to MT destabilization [99,100,101,102]. Many MSAs of natural origin, including taxanes and epothilones (and their derivatives), have been approved for cancer treatment, but their use for neurological alterations has been halted due to their limited brain penetration, poor bioavailability, and/or their potential systemic side effects [103,104,105,106]. Most of these drugs interact with MTs at the taxane-binding site, located in the lumen of the MT in the β-tubulin subunit [107,108,109], counteracting the effects of its GTPase activity [110,111], protecting against MT depolarization and dissolution, and promoting the polarization and structural stability of MTs [112,113,114,115].

Paclitaxel (PTX) is one taxane used in chemotherapy [116,117], that is highly lipophilic and easily crosses the blood–brain barrier (BBB), but it is quickly eliminated from the CNS by p-glycoprotein-mediated transport [118,119,120]. Despite its poor CNS bioavailability, several studies have indicated that PTX could regulate neural shape and function (Figure 2). For instance, PTX promotes axonal elongation/regeneration and reduces glial scar formation in animal models of nerve injury [113,114]. PTX restores axonal transport and reduces the motor phenotype in transgenic mice exhibiting hyperphosphorylated Tau [90], which was linked to MT stabilization because of the increased levels of detyrosinated tubulin [90]. PTX also reduces glutamate-induced neurotoxicity [121] (Figure 2). Despite all these neuroprotective effects, PTX also induces peripheral neuropathy in humans, which is characterized by sensory abnormalities [122], including pain [123] and chemotherapy-induced cognitive impairment [124,125], which has been reproduced in animal models [126] and can also be induced with other taxane-related chemotherapeutic agents such as docetaxel (DTX) [127,128,129] (Figure 2). In rats, PTX treatment induces allodynia that correlates with altered brain activity and connectivity [126], which might be related to the PTX-induced transient encephalopathy documented in humans [130]. Moreover, PTX can induce neuronal death accompanied by the loss of MAP2 and the presence of dystrophic neurites [131] (Figure 2).

Epothilone D (Epo-D), also called BMS-241027, KOS862, dEpoB, and CRND66, is a PTX-derived brain-penetrant MT stabilizer [97], also used in chemotherapy [132,133,134], that prevents MT disassembly by interacting with β-tubulin at the taxane-binding site. This compound is promising to treat neurological diseases for its excellent BBB penetration [112,135] and its poor transport by the p-glycoprotein [97,132]. Epo-D can promote axonal sprouting in injured cortical neurons [136] and facilitate recovery of hind limb function after spinal cord injury in rats [137]. Epo-D improves outcomes in the MPTP-induced mouse model of parkinsonism [138] and in transgenic models of AD [48,111,112,139], by increasing MT density, axonal integrity, and neuronal survival. Epo-D reverts the behavioral alterations in MAP6 knockout mice [77], which correlated with an increase in synapse density and improved synaptic long-term potentiation (LTP) [135]. Moreover, Epo-D attenuates Tau pathology, improves MT density, attenuates axonal dystrophy, improves axonal transport, and enhances cognition in a transgenic mouse model of tauopathy [112,139]. Epo-D also restores normal MT dynamics in conditions of Tau disruption [54,139]. Moreover, Epo-D restores the density of mushroom spines affected by lateral fluid percussion brain injury [140]. Some of these neuroprotective effects have been reproduced with epothilone B (Epo-B) [138,141,142,143]. A collection of studies also report neurological adverse effects of epothilones in animal models [94,140,144]. For instance, Epo-D reduces dendritic arborization [94] and the viability of neurons by affecting mitochondrial transport [145], accelerating disease progression in a transgenic model of ALS [145]. Similar alterations can be induced with Epo-B in cortical and adult sensory neurons [142]. Interestingly, epothilone’s efficacy appears to be more effective in younger rather than aged animals with traumatic brain injury by preventing axonal degeneration processes [140,141,142,143,144,145,146,147,148]. Moreover, epothilones produce more neurotoxic effects in aged animals [140,148]. Thus, specific factors like age can determine the net physiological outcomes of some MT stabilizers. Sex differences in axon diameter, MT density, and resistance to stretch injury [148] could also bias the beneficial effects of MT stabilizers. However, this possibility has not yet been properly assessed.

In contrast to MSAs, certain drugs reduce MT polymerization, such as colchicine, vincristine (VNC), and nocodazole (NOC), and interact with free tubulin subunits, decreasing the concentration of free tubulin available to participate in MT dynamics, thus shifting the balance between polymerized and free subunits toward depolymerization and MT net mass loss [149,150,151,152] and incrementing the proportion of labile MTs, which also impair brain function [55]. NOC can increase MT dynamics in rats after spared nerve injury, improving their cognitive function [153]. Subsequently, describing the characteristics of MT and their endogenous and exogenous modulators, we will describe physiological and pathological conditions that modulate MT configuration. After, we will describe the opposite phenomenon: how changes in MT configuration impact neural network function and morphology.

## 4. Microtubular Reconfiguration during Brain Function

The most convincing evidence of MT modulation associated with specific brain functions and states occurs during the reconfiguration processes underlying learning and memory [55,154,155,156]. For instance, there is an increase in the amount of MTs [154], and MT turnover [55], associated with training in different memory tasks. Memory consolidation is accompanied by increased expression of MAP2 and β-tubulin [155,156,157,158], which is reflected in increased MAP2 immunohistochemical staining [159,160,161]. However, the changes in MAPs expression during learning and memory do not always comprise more protein synthesis, as opposite changes have also been found [162]. For instance, both Tau long isoforms expression and Tau-dependent P13K signaling are decreased during cocaine-associated memory formation, which was abolished by Tau overexpression [162]; when extinction is achieved in this paradigm, Tau long isoforms return to basal levels [162]. It is possible that learning-dependent expression changes in these Tau isoforms drive the dynamic state of neuronal MTs [163]. Indeed, activity-dependent Tau post-translational modifications that could affect their interactions with MTs have also been described [92,164]. As will be reviewed in detail, learning and memory can induce a biphasic change in MT stability (measured by tyrosinated tubulin levels) in a stathmin-dependent manner [55]. Changes in MTs have also been associated with memory deficits [165,166]. For example, social isolation and injuries caused by induced cerebral hypoperfusion result in memory deficits that correlate with decreased levels of α-tubulin and MAP2 [165,166] The cognitive impairment in senescence-accelerated (SAMP10) mice also correlates with a reduction in MAP2 and a simplification of the dendritic arbor [167]. Interestingly, as already mentioned, memory deficits in spared nerve injury rats increased the levels of stable MT, reflected in α-tubulin hyperacetylation, which can be reversed by the MT destabilizer NOC [153]. The cellular mechanisms behind the brain function-induced MT reconfiguration are diverse and will be reviewed next.

## 5. Microtubular Reconfiguration during Neuronal Activity

The generalized increase in neuronal activity, induced by KCl depolarization, is enough to induce MT polymerization, which is blocked by inhibiting action potentials generation with the Na+ channel blocker tetrodotoxin [168]. KCl-induced depolarization increases α-tubulin acetylation [169] and MT entry into dendritic spines [168]. Recently, chemogenetic activation of adult dorsal root ganglion neurons increased MT dynamics through tubulin acetylation, which resulted in axonal growth after nerve injury in vitro [170]. KCl-induced depolarization not only increases neuronal firing [171] but also releases neurotransmitters [172,173], including glutamate, GABA, and glycine [172,173], which modulate MT density [52,174,175,176,177,178]. For instance, activation of glutamate receptors modulates MT function by regulating the expression of MAP2 [52,175,176,177,178] and the upregulation of Tau translation and its accumulation in the somatodendritic compartments [179]. As will be reviewed later, NMDA receptor-dependent synaptic activation increases the proportion of dendritic spines containing dynamic MTs, contributing to spine morphological changes [53,168,180,181,182]. NMDA-dependent MT modulation also occurs during the induction and maintenance of both LTP [168,183] and long-term depression (LTD) [184], which have been related to changes in MT dynamicity in an EB3- and MAP2-dependent manner [184]. Most of the changes in MTs described so far could be explained, or at least partially influenced, by Ca^2+^ entry through NMDA receptors [182]. In vitro studies have shown that MTs polymerize at low Ca^2+^ concentrations, whereas MTs disassemble at increased Ca^2+^ concentrations [185,186,187]. These effects have been hypothesized to be mediated through direct interactions of Ca^2+^ with tubulins or, indirectly, by Ca^2+^-dependent regulators of MT assembly, such as calmodulin [188,189,190,191], by Ca^2+^-dependent modulation of MAPs (i.e., tau) [192] or by drebrin [182]. Similar changes in MTs can be induced by blocking glycinergic receptors, which facilitates tubulin polyglutamylation and alters binding of MAP2 to MTs, which is accompanied by reduced motor protein mobility and cargo delivery into neurites [193]. Reduction in synaptic inhibition has been associated with the induction of hyperexcitable states related to epilepsy [4,174]. Interestingly, the levels of Tyr-tubulin and MT dynamicity are dysregulated in both patients with intractable temporal lobe epilepsy and chronic models of epilepsy [194], while drugs that affect MT stability can either increase (i.e., colchicine) or decrease (i.e., noscapine) epileptiform activity [194]. In vitro experiments have shown that optogenetic neuronal stimulation promotes Tau release, which is reproduced in vivo along with the neuron-to-neuron spreading of this MAP in its pathological forms [195,196,197,198]. Next, we will review the influence of endogenous and exogenous MT modulators on neuronal excitability and morphology [196].

## 6. Changes in Excitability and Synaptic Transmission, and Their Morphological Correlates, Induced by Microtubular Reconfiguration

MT polymerization and depolymerization participate in the clustering and stabilization of ion channels [199], including Na^+^ channels [200], transient receptor potential (TRP) channels [201] and Ca^2+^ channels, and influence their functionality [202,203,204]. Thus, it is expected that endogenous MT stability modulators have diverse effects on neuronal excitability [75]. For instance, the absence of Tau protein makes animals resistant to seizures [205,206], while the reduction of Tau expression alleviates seizure burden and improves survival in some genetic models of epilepsy [206,207,208]. Altogether, these findings indicate that either absence or hyperphosphorylation of Tau produce a hypoexcitable state, which correlates with a reduction in neuronal firing and changes in neurotransmitter release probability [209]. However, this seems to not always be the case, since very young AD transgenic mice, expressing a mutant version of tau and already accumulating hyperphosphorylated Tau protein, do not produce seizure-like activity in the presence of the potassium channel blocker 4-aminopyridine [4]. Tau protein can induce changes in Kv4.2 expression dendrites of CA1 pyramidal cells, which alters their excitability and synaptic plasticity [210], as Tau transgenic animals become older, their neurons exhibit depolarized neuronal resting membrane potentials [207,208,209], increased evoked action potential firing [211,212,213] and are more prone to induced epilepsy [214,215]; a finding consistent with studies demonstrating increased seizure prevalence in patients with AD [191,216,217,218,219,220,221,222]. However, there is a report showing that the Tg4510 Tau mouse model, at similar ages, exhibits neurons with reduced action potential frequency [220,221]. Indeed, the rTg4510 transgenic line, which expresses 14-fold mutant Tau, compared to endogenous Tau, exhibits overall cortical hypoactivity [222]. These diverse changes in firing could be explained by tau-induced modulation of Na^+^ channel function [223,224] in different neuronal types and compartments [224,225,226,227,228]. Another endogenous MT modulator that influences Na^+^ channel function is FGF13 [229]. In fact, Fgf13 knockout mice show markers of MT instability correlating with a reduction of Na^+^ channel presence at the cell membrane, which is mimicked by colchicine in wild-type mice [229]. In contrast, FGF13 overexpression or PTX application results in more Na^+^ channel proteins being inserted into the surface membrane [229]. The latter finding indicates that exogenous MT modulators can affect neuronal activity, as will be reviewed next.

We have just mentioned that PTX promotes Na^+^ channel insertion into the surface membrane [229], which correlates with PTX-induced increased levels of endogenous Nav1.7 mRNA levels and Na^+^ current density [230]. Thus, the changes in ion channels induced by PTX [229,230] could be the cause of the non-convulsive status epilepticus, revealed by EEG, induced by chemotherapeutic administration of PTX [231]. This finding is similar to the slight increase in the frequency and duration of epileptiform activity induced by PTX in vitro [232]. In contrast, MT destabilization with NOC decreased or even abolished epileptiform activity in vitro, which was corroborated in vivo using the maximal dentate activation model [232]. This finding contrasts with the increase of spontaneous seizures during chronic epilepsy induced by colchicine, which is based on the decrease in interneuron firing and the reduction of their inhibitory postsynaptic currents [194]. In contrast, the MT-modulating agent noscapine increased the frequency of action potentials in interneurons and boosted their inhibitory postsynaptic currents, halting the progression of spontaneous seizure during chronic epilepsy [194]. In addition, the MT destabilizer VNC produces a complex effect on excitability since it enhances the excitability of some neurons while reducing it in others [225]. Thus, it appears that MT-modulating agents have heterogeneous and complex net effects depending on the molecule family and neuronal type they are tested on.

One cellular compartment that is highly modulated by MTs and has an enormous impact on neuronal excitability is the proximal region of the axon called axon initial segment (AIS) [221,233], which is a specialized compartment that has a high density of voltage-gated ion channels and generates action potentials [234,235,236,237]. This specialized region consists of the proximal portion along the first 20–40 μm of the axon that extends from the axon hillock to the beginning of the myelin sheath [238]. The AIS has a unique cytoskeletal organization compromising cytoskeletal submembrane networks [239,240,241]. These networks consist of MT bundles coated with a dense submembrane protein network containing ankyrin G (AnkG), βIV-spectrin, and actin filaments [239,242], which serve as scaffolds for ion channel localization and maintenance on the membrane. The AIS cytoskeleton forms a transport barrier between the axon and the somatodendritic membrane [243] and regulates axonal entry of cargoes that require selective transport [244]. The AIS also plays a key role in maintaining the molecular and functional neuronal polarity by controlling membrane diffusion and the polarized trafficking of cytoplasmic proteins toward the axon [244,245]. For example, MAP2 is specifically located in the somatodendritic region and this exclusion from the axonal compartment depends on the assembly of the AIS [241]. The AIS is particularly enriched in voltage-gated sodium (Na^+^) and potassium (K^+^) channels that are required for action potential generation, and the membrane-adaptor protein AnkG is the main component of the AIS scaffold that determines the functional and structural properties of the proteins located at the AIS by recruiting and clustering them in this region [246,247].

The AIS is an essential compartment in the integration of the excitatory and inhibitory postsynaptic potentials into action potential generation [248,249]. Components of the AIS, including the cytoskeleton and ion channels, undergo activity-dependent structural changes that modulate neuronal excitability and maintain steady-state firing rates [234,235,236,237,249]. For example, the AIS is elongated in avian neurons deprived of synaptic inputs for several days [236]. In response to chronic depolarization, the AIS location shifts distally in murine excitatory neurons [233,234,250], but proximally in inhibitory interneurons [251]. Functional defects of the AIS due to cytoskeletal alterations have been reported in cellular models of tauopathies [224,227,228,252]. Moreover, pathogenic Tau acetylation destabilizes the cytoskeletal submembrane protein AnkG and the MTs in the AIS [253], changes the location of the AIS [221,253], precludes activity-dependent change in AIS location [233], and reduces excitability [221]. Similarly, MT destabilization with NOC reproduces most of these effects on AIS and excitability [233]. In contrast, stabilizing MTs with Epo-D restores the cytoskeletal barrier in the AIS and prevents Tau mislocalization [253]. Moreover, PTX prevents pathogenic Tau-induced AIS mislocalization and normalizes neuronal excitability [233]. Considering that neuronal excitability is only a part of the basic processes underlying neural network activity [254], next we will review the changes in synaptic activity induced by endogenous and exogenous MT modulators.

At the presynaptic compartments, the cytoskeleton enters deep into presynaptic terminal swellings and partially colocalizes with a subset of synaptic vesicles (SVs) [255]. Functionally, it is thought that this interaction regulates neurotransmitter release [255,256], mediates endocytosis of SVs [257,258,259], and promotes the recovery of synaptic responses from activity-dependent short-term depression [260,261] via fast SV replenishment [262], clearance of used SVs from release sites [263,264,265,266], and transport of mitochondria and presynaptic elements [266,267]. Early electron microscopy studies at the frog neuromuscular junction reported that MTs anchoring SVs are directed toward active zones [268,269,270]. Likewise, at the Drosophila NMJ, the MT-associated protein Futsch [271] links MTs to AZs, thereby supporting neurotransmitter release [256]. At the calyx of Held in adult cats, MTs are observed in presynaptic terminal swellings, but not in the SV pool [272]. Depolymerization of MTs with NOC impairs long-distance SV movements between presynaptic swellings [273]. Dynamic MTs preferentially growing in presynaptic boutons show biased directionality in that they are almost always oriented toward the distal tip of the axon, which can be modulated by neuronal activity [23]. Silencing γ-tubulin expression reduces presynaptic MT nucleation, SV interbouton transport and regulates evoked SV exocytosis [23]. Dynamic MTs are enriched at *en passant* boutons and allow for the targeted delivery and unloading of SV precursors by the kinesin-3 motor KIF1A [274]. In *en passant* boutons, presynaptic dynamic MTs are nucleated upon neuronal activity and are critical for adjusting activity-evoked neurotransmitter release by providing paths for interbouton bidirectional transport of SVs, which is a rate-limiting step in SV unloading and exocytosis at release sites [23,274]. MT nucleation preferentially occurs at excitatory boutons in hippocampal slices from neonatal mice [23]. Dynamic MTs may be directly regulating Ca^2+^ handling at terminals through the interaction of EB1/3 with endoplasmic reticulum Ca^2+^ sensors [275,276,277]. In simultaneous presynaptic and postsynaptic action potential recordings, depolymerization of MTs impaired the fidelity of high-frequency neurotransmission at the calyx of Held presynaptic terminals [24]. Thus, it would be expected that modifications in MTs would have a major impact in synaptic transmission.

As shown for neuronal excitability, changes in Tau also affect basal synaptic activity [209,211,212,213,278,279]. As mentioned, Tau knockout changes synaptic release probability which is reflected in an increase in paired-pulse facilitation [209]. In contrast, the overexpression of mutated human Tau increases spontaneous excitatory postsynaptic currents (EPSPs) [211,212,213], increases glutamate release, and decreases glutamate reuptake [278,279], while also decreasing paired-pulse facilitation [280,281,282,283]. MT influence on synaptic transmission has been observed after the depletion of the MT-severing protein spastin, which produces longer MTs with increased tubulin polyglutamylation leading to a lower frequency of miniature EPSCs [284]. Beyond these effects of endogenous MT modulators on synaptic transmission, there is extensive evidence of the effects of pharmacological MT modulators on this phenomenon. For instance, MT stabilization with PTX increases the frequency of miniature EPSCs and reduces the paired-pulse facilitation of evoked EPSCs, which is reversed by the NMDA receptor antagonist 2-amino-5-phosphonopentanoic acid [285]. This is similar to the increase in amplitude and frequency of miniature inhibitory postsynaptic currents induced by noscapine [194]. These observations contrast with the effects observed by MT destabilization with colchicine, which reduces the amplitude and mIPSCs [194], similarly to the reduction in synaptic transmission recovery in the presence of vinblastine at the calyx of Held presynaptic terminals [24]. As mentioned, in these highly active synapses the presynaptic MTs play important roles in SV cycling and mitochondrial anchoring [24,273,286]. Although we will later discuss the influence of MTs on dendritic spines, it is important to conclude this section by indicating that NMDA and AMPA receptors trafficking, depends heavily on stable MT-mediated transport [57,287,288]. For instance, NOC or colchicine agents inhibited NMDA receptor-mediated function and, thus, synaptic currents in an MT-dependent manner [287]. The effect of MT depolymerizers, which is most prominent in NR2B subunit-containing NMDA receptors, was blocked by cellular knockdown of the kinesin motor protein KIF17, which transports NR2B-containing vesicles along MTs in neuronal dendrites. Moreover, immunocytochemical studies show that MT depolymerizers decreased the number of surfaces NR2B subunits on dendrites, all of which were reversed by brain-derived neurotrophic factor (BDNF) and PTX [289]. To further support the role of MTs in synaptic receptors, memory consolidation regulates, in a stathmin-dependent manner, the transport of the GluA2 subunit of the AMPA receptor, resulting in increased GluA2 at synaptic sites, which promotes long-term memory [56]. The complex relationship between MTs and memory will be described next.

## 7. Changes in Long-Term Synaptic Plasticity and in Dendritic Spines Induced by Microtubular Modulation

Endogenous and pharmacological MT modulation has a major impact on learning and memory [55,56]. To understand this impact, we will first review the effects of endogenous and pharmacological MT modulation on two of its most likely underlying cellular mechanisms, namely long-term synaptic plasticity [290] and dendritic spine reconfiguration [1,53,168,182,187,291,292]. There is extensive evidence that changes in MTs strongly affect long-term plasticity [55,56,293,294]. For instance, LTP is highly dependent on normal Tau function, in a very narrow homeostatic range, since this process is abolished by either Tau knockout [295] or the overexpression of pathological forms of Tau [278,280,281,282,283,296]. Pathological Tau constructs reduce LTP in CA3-CA1 connection [296,297], while LTD is altered by the presence of hyperphosphorylated Tau due to changes in NMDA receptor activity [164,298]. In contrast, it is also reported that young transgenic mice expressing a mutant form of Tau exhibit an increase in LTP [299], while LTD can be inhibited by very low concentrations of oligomeric Tau [300]. Another MT modulator that plays a major role in LTP regulation is stathmin, a protein that binds tubulin and inhibits MT polymerization [56,293]. Mice lacking the stathmin4A isoform or its non-phosphorylatable mutant exhibit deficits in LTP generation in the cortico-amygdala, thalamo-amygdala, and the perforant path to the dentate gyrus synapses, but not at the Schaffer collaterals to CA1 [56,293]. Spastin depletion also reduces LTP [284]. Altogether, these findings indicate that normal MT function is required for the induction of synaptic potentiation, while the use of exogenous MT modulators reveals a similar scenario. Slices treated with NOC cannot maintain post-tetanic potentiation or LTP [53,55,301], although other authors have not found this decline in the presence of VNC [302]. However, synaptic potentiation in the presence of VNC becomes sensitive to the co-application of protein synthesis inhibitors [301]. Similarly, MT stabilization with PTX also reduces LTP in the cortico-amygdala and CA3-CA1 synapses [149,293]. However, Epo-D can reestablish LTP in animals lacking MAP6, which were unable to induce such potentiation, at the CA3-CA1 synapse [135]. As for LTP, LTD is highly sensitive to MT modulation [303,304]. Hippocampal LTD deficit is common in Tau knockouts [303,304] or mice with a reduced expression of Tau [304]. Nonetheless, Epo-D does not affect LTD induction at the Schaffer collaterals reaching the CA1 [135].

Dendritic spines are small micrometer-sized specialized protrusions of the membrane that decorate dendritic branches [305] and act as dynamic microcompartments to restrict and amplify excitatory signaling [306] and whose plasticity has been associated with a variety of neural functions, including learning and memory [166,181]. Spine shape and function were classically considered to be mainly determined by actin filaments, which are highly enriched in spines [307]. However, now it is well accepted that MTs are determinant in the development, maintenance, plasticity, and degeneration of spines [53,308,309,310,311,312]. For instance, pR5 transgenic mice, which overexpress hyperphosphorylated Tau, show significant changes in dendritic morphology in CA1 pyramidal neurons [313]. Moreover, dynamic MTs appear to regulate dendritic spine morphology and synaptic plasticity [53] and promote NMDA receptor and Ca^2+^-dependent spine enlargement [181] by continuously invading all types of spines (mushroom, stubby and thin, as well as filopodia). Large spines consistently exhibit transient and activity-dependent invasion of MTs [168,314], which mainly relies on Ca^2+^ [168,180,182], membrane depolarization [168] and the interaction of MTs with actin through drebrin [178,315,316]. MT invasion of spines is also highly dependent on the end-binding protein 3 (EB3) [53]. In fact, inhibition of MT growth by depletion of EB3 caused the specific loss of mushroom-headed spines and increased the percentage of filopodia [53,314]. As expected, EB3 overexpression causes the increase of mushroom-headed spines [53,316]. Moreover, EB3 overexpression reverses the deficiency of mushroom spines in AD transgenic mice [317].

Spine invasion by MTs is a transient event [53,168,181,182,184,291,292] that occurs under physiological conditions [53,168,312] but that is exacerbated with stimulations that induce LTP [168,182,318,319], with membrane depolarization [168] or with the application of BDNF [320]. NMDA receptor-dependent synaptic activation increased the proportion of dendritic spines containing dynamic MTs, which then contributed to spine enlargement [53,168,180,182]. On the other hand, inhibition of NMDA receptor activity reduced MT invasion of spines [181]. In contrast, stimulation of both synaptic and extrasynaptic NMDA receptors by bath application of NMDA results in a loss of MT dynamics in dendrites and spines [184], inducing LTD [184], which requires removing EB3 from the growing MTs in a Ca^2+^-dependent manner [184]. LTP and high KCl-induced increase of dendritic spines containing MTs were completely abolished by inhibiting the firing of action potentials with TTX [168]. Long-term treatment of hippocampal cultures with BDNF increases spine number, which is further increased by the presence of the MT-stabilizing agent PTX [314]. In contrast, disruption of MTs with NOC blocks the spine-promoting effect of BDNF [314]. Importantly, MT entry into spines was increased after the transient stimulation with KCl, and this increase was blocked by treatment with TTX, indicating that MT dynamics in neurons are changing in an action potential-dependent manner [168].

Neuronal MAPs, such as MAP2 and Tau are also involved in regulating MT dynamics and interactions in dendritic spines [321,322]. MAP2 binds along the length of MTs but is also associated with actin in dendritic spines [319] and interacts with the NMDA receptor subunits NR2A and NR2B [322]. Similar to MAP2, Tau interacts with PSD-95, which in turn regulates NMDA receptor through the tyrosine kinase Fyn [164]. Therefore, NMDA receptor-mediated Tau phosphorylation at specific residues results in the weakening of the tau–PSD95–Fyn interaction, regulating postsynaptic plasticity [5,164]. Dynamic MT spine invasion regulates spine morphology [93,323,324], synaptic plasticity [168,183], the recycling of endosomes containing AMPA receptors into spines from the dendritic shaft [325] and the content of PSD95 in the spines [320]. Syntaptotagmin4-containing vesicles are also transported by polymerizing MTs into spine heads, where they subsequently undergo exocytosis [292]. Thus, it is not surprising that shortening of dendritic spines and changes in spine shape (i.e., shift from mushroom to stubby spines) appear to be relevant indicators of the progression of cognitive deficits [92,323,324] and that these and other alterations of spines have been reported in different brain pathologies such as autism spectrum disorders, schizophrenia, and fragile X syndrome [69,326,327].

Exogenous MT modulators have a huge impact on spine shape and function. For instance, NOC causes spine loss without changing spine morphology [48]. In another study, NOC abolished EB3 accumulation at MTs and reduced the number of mushroom spines while increasing the number of filopodia [53]. In addition, NOC suppresses the spine recovery induced by a gamma-secretase inhibitor in AD transgenic mice [48]. NOC also reversed the memory-induced increase in MAP2-associated MTs, reducing dendritic spine density, and impairing memory formation. The effects of NOC on MT turnover were prevented by PTX and BDNF, which restored dendritic spine density and memory formation [55]. Similarly, NOC inhibits the BDNF-induced increase in the spine density, yet [314] it also partially restores the number of mushroom spines and spine density in AD transgenic neurons, possibly by promoting dynamic MT entry into the spines [48]. Similarly, Epo-D also reverses Aβ-induced spine loss [328], which would appear to be counterintuitive, as Epo-D reduces mushroom spines in wild-type slice cultures [328]. Thus, it is possible that Epo-D prevents MT disassembly in AD transgenic neurons and maintains spine morphology; however, it also inhibits MT dynamics required for spine maturation in normal mice [328]. In fact, other groups have found that while Epo-D alters dendritic spine length, density, and morphology [136,329,330], it can also reduce spine length and increase the density of mushroom spines after fluid percussion injury [140]. PTX also reduces dendritic spine density, which is mitigated in Tau knockout neurons [331]. Although PTX alters the dynamics of dendritic spines [332], it also prevents the reduction of spine density and memory alterations induced by NOC [55] and exacerbates the BDNF-induced increase in the spine density [314].

## 8. Changes in Learning and Memory Induced by Microtubular Modulation

Learning and memory require the proper representations of experiences that become imprinted in neuronal circuits during memory consolidation [290], which would involve functional and morphological changes that depend on MT function. Thus, it is not surprising that one of the most challenging side-effects of MSA-based chemotherapy is learning and memory impairments [130,333,334,335,336]. However, as will be reviewed next, MT instability or stability can either promote or impede learning and memory in a state-dependent manner [56,57]. The clearest example of this dynamic relationship was provided by Uchida et al. (2014) and Yousefzadeh et al., (2021) [56,337] who found that learning and memory cause biphasic changes in MTs. In the early phase of the process, stathmin dephosphorylation enhances MT-instability, whereas in the late phase these processes are reversed and a hyperstable MT state is achieved during context-fear memory [56]. As expected, PTX administration immediately following the training precludes memory formation [56,337] but increases memory when applied during its maintenance [56,337].

The complex interaction between MTs and learning and memory is well exemplified by the diverse actions that Tau exerts on MT. For instance, the expression of Tau mutant variants in *Drosophila* not only alters the cytoskeleton at the synaptic terminals but also modifies neuronal activity patterns and memory consolidation [338]. Similarly, normal Tau overexpression results in learning and memory deficits [339,340]. Moreover, old transgenic mice expressing mutant forms of Tau also exhibit deteriorated memory [54,116,143] and disrupted LTP [278,280,281,282,283,296]. However, young transgenic mice expressing a mutant form of Tau exhibit improved memory [299], which correlates with increased LTP [299]. Changes in other MT regulators also affect learning and memory in a complex manner. In the case of stathmin, its knockout impaired memory and LTP [293], while the expression of the non-phosphorylatable and constitutively active form produces similar effects [58]. Other examples of learning and memory alterations induced by the reduction of MT stabilizers have been found in mice lacking FGF13 [75], spastin [286], CRTC1 [341], or KIF21B [342].

The effects of exogenous modulators of MT stability on learning and memory are also complex. Epo-D may have some neuroprotective effects on this phenomenon since its application improves cognitive performance in Tau transgenic mice [54,112,139,343] by increasing MT density and axonal integrity and decreasing hyperdynamic MTs [54,112,139,343]. Similarly, EpoD treatment has beneficial effects on APP/PS1 double-transgenic mice, improving their axonal transport of mitochondria-associated with enhanced motor and spatial memory [344]. However, Epo-D induces an alteration in reversal learning (crossover) in these animals [344]. Similarly, PTX can prevent traumatic brain injury-induced deficits in memory [345], which is due to the prevention of structural injury and hypometabolism [345]. PTX also prevents the memory impairment induced by NOC [55]. However, as already mentioned, PTX can have a deleterious effect on learning and memory [56], which has been observed by many groups [143,153,346,347,348,349,350,351] suggesting that state-dependent physiological MT dynamics, rather than an overall shift to stabilization, is important for learning and memory. Thus, since a moderate stabilization of MTs may be protective, prevention of MT dynamics can have a detrimental effect on these plastic phenomena [154,346,347,348,349,350,351,352,353]. In fact, under conditions in which memory was not impaired, PTX treatment impaired learning of new rules [59]. PTX-induced memory impairment, which can be prevented by lithium [351], has been related to a decrease in LTP and [154], neurogenesis [351], an increased number of TUNEL-positive neurons, increased expression of TNF-α and IL-1β [352] and can be reduced by the TNF-α synthesis inhibitor thalidomide [353], which indicates that this phenomenon could be related to neuroinflammation [353]. However, it is important to notice that others have found that PTX does not induce brain inflammation, as measured by cytokine analysis, which correlates with the lack of effect of aspirin on PTX-induced memory alteration [346]. PTX-induced memory impairment has also been associated with a reduction in dendritic length and complexity [354], which is also reverted by lithium [354]. This impairment has also been related to a reduction in cell proliferation [350,351]. As described for PTX, DTX also induces alterations in memory [347,348,355] which, in this case, is related to elevated neural autophagy and astrocytic activation [352] and is reversed by rolipram administration [50,348].

The clearest evidence that MT destabilization affects memory is provided by extensive demonstrations that colchicine affects learning and memory in an MT-dependent manner [355,356,357,358,359]. However, it should be acknowledged that, at high doses, colchicine can induce cell death [360,361,362,363,364,365,366]. Although MT destabilization with NOC administration immediately following training could promote learning [55,336], NOC also can inhibit memory formation if administered before learning [54] and even reduce memory retrieval if administered at late phases of learning [55,336]. However, NOC can also prevent the memory deficits induced by spared nerve injury [152].

In addition to the modulation of encoding and retrieval, some effects of MT stabilizers could also be interpreted as impairments on cognitive flexibility, which comprehends adaptative changes in the behavioral output in response to modifications of the rules of the task [367,368]. For instance, weekly administration of Epo-D not only prevented APP/PS1 mice from exhibiting retrieval impairments in the water Morris maze [344], but also caused less extinction, that is expected when the platform is removed for several weeks [344]. Moreover, the injection of PTX in the dorsolateral striatum, after the learning of a specific target duration in a temporal learning paradigm, prevents the acquisition of a new target duration but strengthens the recall of the old one [337]. This rigid behavior also accounts for reduced cognitive flexibility [367,368]. An elegant work showed that PTX has negative effects specifically on reversal learning, sparing prior training with simple discrimination of pairs of odorants and the learning of new pairs of odorants [350]. Moreover, a single dose of PTX to cancer patients has been associated with confusion, and word recollection impairments [130,336]. On the other hand, MT destabilization can also impair cognitive flexibility. *Stat4A* mice have reduced reversal learning in the Morris water maze [57], which comprehends the relocation of the hidden platform after the training with the original location [57]. These effects of MT modulation on cognitive flexibility could be due to the impairments induced by MT stabilizers and destabilizers on brain connectivity [126]. MT stabilizers-induced neuropathic pain [122,123] could also reduce cognitive flexibility as observed in some pain models [369].

Overall, it appears that either endogenous homeostatic and pathological MTs modulation, or exogenous pharmacological modulation of MT dynamicity, encompasses favorable and detrimental effects in brain shape and function. Some factors like age, sex, brain region, neuronal type, learning, activity-dependent processes, and behavioral tasks influence the dynamic state of neuronal MTs and bias the effects of some stabilizing and destabilizing drugs. This could be due to the exquisitely regulated allostasis that MTs exhibit to respond effectively to specific neuronal demands in their cytosolic microenvironment. Further research must be focused on the underlying mechanisms of MT-dependent processes that modify behavior with the more temporal and temporal resolution, considering age, sex, and strain as key factors. Thus, some basic gaps to fill are the sex, brain region, and neuronal type dependence of the MT modulation and its effects on brain shape and function. It would be relevant to also investigate the protein-protein interactions and post-translational modifications occurring during MT dynamic response to neuronal function.

## Figures and Tables

**Figure 1 cells-11-00923-f001:**
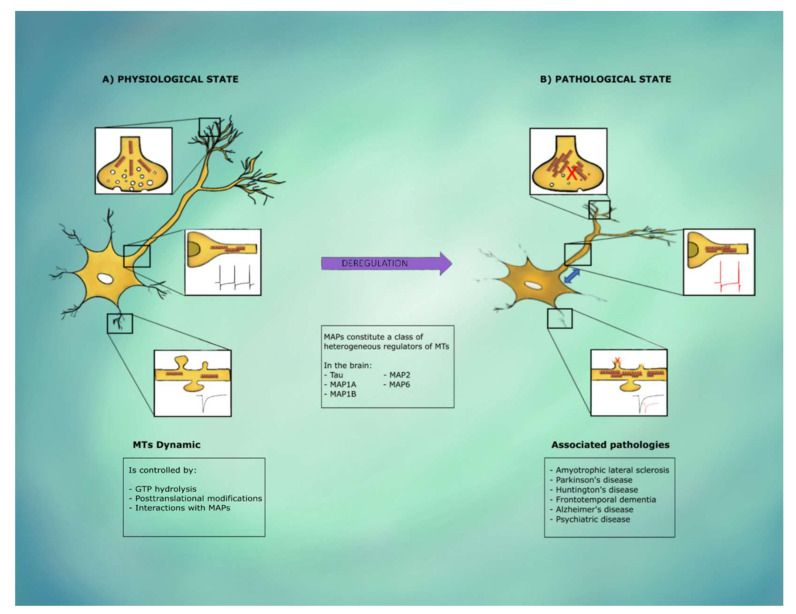
Microtubules (MTs) function in different neuronal compartments. In physiological conditions (**A**), MTs allow synaptic vesicles’ transport and recycling for proper synaptic transmission and plasticity. MTs contribute to the organization of the axon initial segment for action potential initiation and plasticity of intrinsic excitability. MTs are also capable of invading dendritic spines in an activity-dependent manner, for cargo delivery and postsynaptic plasticity regulation. However, when MTs dynamics and stability processes exceed the normal homeostatic range, under pathological conditions (**B**), there are MTs-dependent alterations in synaptic transmission due to inefficient synaptic vesicles’ transport, aberrant firing activity due to the relocation of the axon initial segment, loss of postsynaptic plasticity and alterations in dendritic spines due to reduced responsivity of MTs located in the dendritic shaft.

**Figure 2 cells-11-00923-f002:**
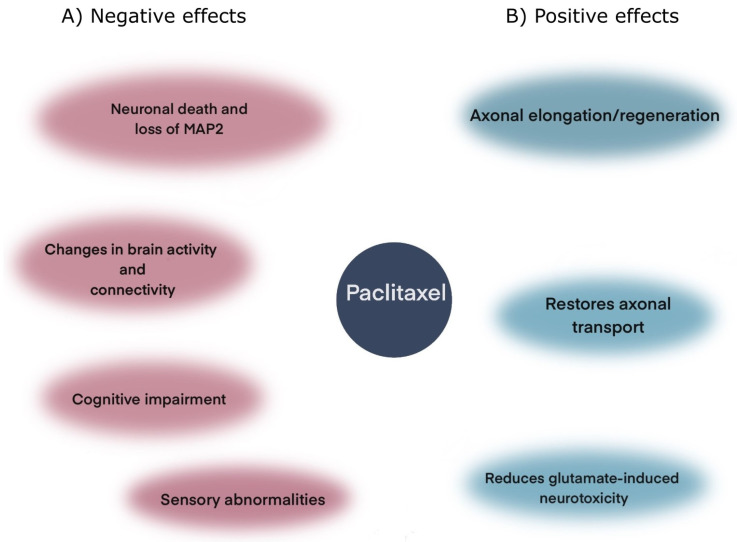
Differential effects caused by Paclitaxel. Paclitaxel is a microtubules stabilizer used in chemotherapy, which is capable of exerting both negative and positive effects on neural function and shape. (**A**) The most common negative effects associated with PTX are neuronal death, cognitive impairment, and sensory abnormalities (i.e., allodynia), that correlate with changes in brain activity and connectivity. (**B**) On the other hand, some positive effects of PTX are the enhancement of regeneration and elongation of axonal processes, restoration of axonal transport, and the reduction of glutamate-induced neurotoxicity.

## Data Availability

Not applicable.

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
