# Peer review of "Microtubules as Regulators of Neural Network Shape and Function: Focus on Excitability, Plasticity and Memory"

_cells, 2022, doi:10.3390/cells11060923_

Round 1

Reviewer 1 Report

This review written by Peña-Ortega and colleagues describes the role of MTs as regulators of brain networks from a structural and functional point of view focusing on synaptic plasticity and memory. This review is overall well written and exhaustive in the description of MTs, however I have some consideration.

I strongly suggest the authors to add explanatory figures, such as a cartoon of MTs structure and localization in neurons to facilitate the understanding of the description in the text. Because this is a review, and it describes many aspects of MTs, it needs to be accompanied by at least a couple of descriptive figures. Perhaps it would be interesting to see a WT condition compared to a pathological state in which MTs are affected and their functions might be compromised, especially regarding dendritic spine morphology.

Another aspect that might be interesting to discuss in this review, since the focus is the role of MTs in learning and memory processes, are evidence about MTs dysregulations and altered performances in cognitive tasks in animal models, to better correlate possible MTs alterations with cognitive flexibility.

A short paragraph with concluding remarks might be useful to summarize the whole content of this review and clarify a few important take home messages and perhaps a comment about what future studies should address.

Line 78: the authors write “transport [] (Vale et al., 1985)”. The reference is outside the squared brackets, that indeed are empty. Please correct.

Author Response

Reviewer 1

I strongly suggest the authors to add explanatory figures, such as a cartoon of MTs structure and localization in neurons to facilitate the understanding of the description in the text. Because this is a review, and it describes many aspects of MTs, it needs to be accompanied by at least a couple of descriptive figures. Perhaps it would be interesting to see a WT condition compared to a pathological state in which MTs are affected and their functions might be compromised, especially regarding dendritic spine morphology.

Answer: The figures requested by the reviewer, as well as the other reviewer, have been included.

Another aspect that might be interesting to discuss in this review, since the focus is the role of MTs in learning and memory processes, are evidence about MTs dysregulations and altered performances in cognitive tasks in animal models, to better correlate possible MTs alterations with cognitive flexibility.

Answer: The required information has been included.

A short paragraph with concluding remarks might be useful to summarize the whole content of this review and clarify a few important take home messages and perhaps a comment about what future studies should address.

Answer: The paragraph with the concluding remarks has been included.

Line 78: the authors write “transport [] (Vale et al., 1985)”. The reference is outside the squared brackets, that indeed are empty. Please correct.

Answer: The reference has been corrected.

Reviewer 2 Report

Summary: The authors have reviewed literature to provide insights into how microtubles play an important role in health and disease to determine outcomes on neuronal excitability, synaptic plasticity and the formation and expression of memory. There are some needs to refine the language and content as outlined below:

Major points:

1) The paclitaxel paragraph from line 132 onwards is one of the best written one which deserves a model figure to describe the myriad roles of paclitaxel depending on the environment.

2) In the paragraph on Epo-D starting line 150, it is becoming very important to consider how there can be differences that can be attested to the age as well as the sex of the model organisms. Both in this paragraph and other paragraphs, these sex-specific effects need to be elaborated. At the very least, there is need (according to revised NIH rules and regulations) to consider sex as a variable and more recently, diversity as a key factor influencing the signaling mechanisms. Along the lines of diversity, recent literature has looked at whether the studies being reviewed/included have taken into account the strain - C57 or DBA etc.

3) The differences of Tau expression in sentences associated with line 191 can be attributed to consolidation vs extinction - this needs to be discussed so that there is appreciation of how the signaling needs to be studied in conjunction with the outcome - different types of memory processes.

4) The paragraph ends abruptly at 569, without an overall summary paragraph or a take-home message. The review requires a model figure summing the explanations to really highlight the importance of the topic elaborated here.

Minor points:

1) Line 59 - change "undergoe" to"undergo"

2) Line 179 - Use "Subsequently" instead of "After"

3) Line 183 - replace "to" by "with"

4) the phrase "while ...." in line 197 does not make any sense with the rest of the sentence, reword.

5) Line 488 - change "los" to "loss"

Author Response

Reviewer 2

Major points:

1) The paclitaxel paragraph from line 132 onwards is one of the best written one which deserves a model figure to describe the myriad roles of paclitaxel depending on the environment.

Answer: The specific figure requested by the reviewer, as well as other figures, have been included.

2) In the paragraph on Epo-D starting line 150, it is becoming very important to consider how there can be differences that can be attested to the age as well as the sex of the model organisms. Both in this paragraph and other paragraphs, these sex-specific effects need to be elaborated. At the very least, there is need (according to revised NIH rules and regulations) to consider sex as a variable and more recently, diversity as a key factor influencing the signaling mechanisms. Along the lines of diversity, recent literature has looked at whether the studies being reviewed/included have taken into account the strain - C57 or DBA etc.

Answer: We have included the requested information in those cases in which it was available. Of notice, not so many of the included papers provided the requested information (which was mentioned at the end of the review).

3) The differences of Tau expression in sentences associated with line 191 can be attributed to consolidation vs extinction - this needs to be discussed so that there is appreciation of how the signaling needs to be studied in conjunction with the outcome - different types of memory processes.

Answer: We have included the requested information.

4) The paragraph ends abruptly at 569, without an overall summary paragraph or a take-home message. The review requires a model figure summing the explanations to really highlight the importance of the topic elaborated here.

Answer: The paragraph with the concluding remarks has been included.

Minor points:

1) Line 59 - change "undergoe" to"undergo"

Answer: We have changed the word as suggested.

2) Line 179 - Use "Subsequently" instead of "After"

Answer: We have changed the word as suggested.

3) Line 183 - replace "to" by "with"

Answer: We have changed the word as suggested.

4) the phrase "while ...." in line 197 does not make any sense with the rest of the sentence, reword.

Answer: We have eliminated the sentence and integrated the information to the previous one.

5) Line 488 - change "los" to "loss"

Answer: We have changed the word as suggested.

Round 2

Reviewer 2 Report

The authors have revised the manuscript satisfactorily and the review is suitable for publication.